

**Impact of the Green Light Program on haze pollution in**
**the North China Plain, China**
Xin Long[1,3,7], Xuexi Tie[1,2,3,4,5*], Jiamao Zhou[3], Wenting Dai[3], Xueke Li[6], Tian
Feng[1], Guohui Li[1,3], Junji Cao[1,3], and Zhisheng An[1]
[1]State Key Laboratory of Loess and Quaternary Geology, SKLLQG, Institute of Earth
Environment, Chinese Academy of Sciences, Xi'an 710061, China
[2]Center for Excellence in Urban Atmospheric Environment, Institute of Urban Environment,
Chinese Academy of Sciences, Xiamen 361021, China
[3]Key Laboratory of Aerosol Chemistry and Physics, Institute of Earth Environment, Chinese
Academy of Sciences, Xi'an 710061, China
[4]Shanghai Key Laboratory of Meteorology and Health, Shanghai, 200030, China
[5]National Center for Atmospheric Research, Boulder, CO 80303, USA
[6]Department of Geography, University of Connecticut, Storrs, Mansfield, CT 06269, USA
[7]School of Environment Science and Engineering, Southern University of Science and
Technology, Shenzhen 518055, China
*Correspondence to:* Xuexi Tie (email: xxtie@ucar.edu)





**Abstract.** As the world's largest developing country, China undergoes the ever-increasing
demand for electricity during the past few decades. In 1996, China launched the Green Lights
Program (GLP), which becomes a national energy conservation activity for saving lighting
electricity, as well as an effective reduction of the coal consumption for power generation.
Despite of the great success of the GLP, its effects on haze pollution have not been
investigated and well understood. This study focused to assess the potential coal-saving
induced by the GLP and to estimate the consequent improvements of the haze pollutions in the
North China Plain (NCP), because severe haze pollutions often occur in the NCP and a large
amount of power plants locate in this region. The estimated potential coal-saving induced by
the GLP can reach a massive value of 120–323 million tons, accounting for 6.7–18.0% of the
total coal consumption for thermal power generation in China. In December 2015, there was a
massive potential emission reduction of air pollutants from thermal power generation in the
NCP, which was estimated to be 20.0–53.8 Gg for NOx and 6.9–18.7 Gg for $SO_2$. The
potential emission reductions induced by the GLP played important roles in the haze formation,
because the NOx and $SO_2$ are important precursors for the formation of particles. To assess the
impact of the GLP on haze pollution, sensitive studies were conducted by applying a regional
chemical/dynamical model (WRF-CHEM). The model results suggest that in the lower limit
case of emission reduction, the $PM_{2.5}$ concentration decreases by 2–5 $\mu g \ m^{-3}$ in large areas of
the NCP. In the upper limit case of emission reduction, there was much more remarkable
decrease in $PM_{2.5}$ concentration (4–10 $\mu g \ m^{-3}$). This study is a good example to illustrate that
scientific innovation can induce important benefits on environment issues, such as haze
pollution.

**Keywords:** Green Light Programs; thermal power plants; Haze in NCP; WRF-CHEM





## 1 Introduction


As the world's largest developing country, China undergoes the ever-increasing demand for
electricity during the past few decades. Artificial lighting is an important part of China's
energy consumption, accounting for a quite stable share of about 10–14% of the total
electricity consumption (Lv and Lv, 2012; Zheng et al., 2016). Also, the lighting demand in
China is predicted to increase continuously, with a projected average annual growth rate of 4.3%
from 2002 to 2020 (Liu, 2009). With principal objective of alleviating shortage of electricity,
China has launched the Green Lights Program (GLP) in 1996, with the core of aiming at
replacing low-efficiency lighting lamps by high-efficiency ones. Since then, the GLP has
become a national energy conservation activity for saving lighting electricity, and has been
highlighted continuously in the nation's $9^{th}$–$12^{th}$ Five-Year Plan (1996–2015) (Lin, 1999).
With the object of providing high-quality efficient lighting products, the GLP is undoubtedly a
useful electricity-saving measurement. Nonetheless, driven by the accelerated economic
increase, the thermal power electricity has experienced an ever-increasing trend in the past
decades, as well as the associated coal consumption for thermal power generation. Thermal
power generation is the primary electricity source in China, contributing to about 72–78% out
of the total electricity (NBS, 2000–2016). In 2015, the coal consumption for thermal power
generation in China raise to a very massive value of about 1.8 billion tons, which is 3.2 times
of that in 2000. Simultaneously, the coal consumption for thermal power generation is 2.7
times of that in the USA in 2015, which is reported to be 670 million tons
(https://www.eia.gov/totalenergy/data/browser/, last accessed on 20 December, 2018).
Due to the significant use of coal, thermal power generation is one of the dominant emission
contributors to anthropogenic air pollutants in China (Tie and Cao, 2010; Wang and Hao, 2012;





Wang et al., 2015b). The power sector contributes significantly to air pollutants of the nitrogen
oxides (NOx), the sulfur dioxide ($SO_2$), and the particulate mater (PM) (Zhao et al., 2013;
Huang et al., 2016). The pollutants of $SO_2$ and NOx are the precursors of secondary pollutant
of ozone ($O_3$), and secondary aerosols (Seinfeld et al., 1998; Laurent et al., 2014). It is also
reported that emission from power sector is a major contributor to particulate sulfate, and
nitrate (Zhang et al., 2012). The emissions from thermal power generation in China can also
transport to a long distance, causing regional/global air pollutions (Tie et al., 2001; Huang et
al., 2016). Considering the important contributions to air pollutants, controlling emissions from
thermal power generation is a vital strategy for the improvement of air quality in China.
Distinguished from the ever-increasing trend of thermal power electricity and associated coal
consumption, the increase trends of $SO_2$ and NOx emissions from thermal power generation
are curbed and even change to decrease (Liu et al., 2015). This is caused by the famous
nation-wide project of utilizing emission control facilities during 2005 to 2015, such as
installing flue-gas desulfurization/denitrification systems and optimizing the generation fleet
mix (Liu et al., 2015; Huang et al., 2016). Given the technological changes that have occurred
in the power sector, the air pollutant emissions from power generation have been significantly
reduced. However, the thermal power generation is still identified to be with massive air
pollutant emissions, involving 5.1 million tons of NOx, 4.0 million tons for $SO_2$, and 0.8
millions tons of PM in 2015. Under high standards of ultra-low emission power units, the
staggering total amount of coal consumption becomes a vital challenge for emission control
from thermal power generation.
With ambitious and comprehensive efforts, the success of the GLP resulted in about 59 billion
kWh of accumulated electricity savings from 1996 to2005 (SCIO, 2006), and about 14.4

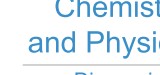



billion kWh of annual electricity savings from 2006 to 2010 (Lv and Lv, 2012). It is reported
that the GLP has produced climate benefit for environment, reducing 17 million tons of $CO_2$
and 530 thousand tons of $SO_2$ emissions from 1996 to 2005 (Guo and Pachauri, 2017).
Coordinate with the effectiveness of the GLP on energy saving, the effects of power generation
or coal-saving on air quality are elaborated in previous studies (Liu et al., 2015; Huang et al.,
2016; Hu et al., 2016).
However, few studies have been so far dedicated to estimate the effectiveness of the GLP in
controlling air pollution on a regional scale, especially in North China Plain (NCP). In the NCP,
the thermal power plants are very densely distributed, resulting in massive emissions of air
pollutants (Liu et al., 2015). As a result, the GLP could produces significant energy-saving and
reduces the associated air pollutant emissions from thermal power generation. Although the
GLP is under the strong and sustained government commitment, however, there is no built-in
mechanism for monitoring the GLP and without regularly issued official program assessment
reports (Guo and Pachauri, 2017). During the past decades, the Chinese government has
published only one report regarding the performance of the GLP (NDRC, 2005). There are
several articles and books for summarizing the GLP from time to time by the Energy Research
Institute under Chinas' NDRC, providing additional information for assessments (Yu and Zhou,
2001; Liu, 2006; Liu and Zhao, 2011; Liu, 2012; Lv and Lv, 2012; Gao and Zheng, 2016).
Previous studies do not well investigate the effects of the GLP on air pollution, such as the
resultant of emission reductions of air pollutants, or the consequent effects on haze pollution.
In the present study, we quantified the effect of the GLP on the haze pollution in the NCP, a
severe air polluted region in China. The study included satellite measurements and numerical
model studies (WRF-CHEM). We first investigated the lighting coal consumption and



resultant coal-saving induced by the GLP utilizing the satellite nighttime lights (NTL) data
(Elvidge et al., 2009), which has been widely used to estimate the consumption of energy and
electricity (He et al., 2013; Huang et al., 2014). Then we evaluated the potential emission
reductions and resultant effects on air pollution in the NCP using the WRF-CHEM model. This
study provided an overall perspective on gaps of the unevaluated potential benefits to haze
pollution induced by the GLP, which can inspire more macroscopic and interdisciplinary
analysis in long-term national activities based on NTL datasets. We summarized the data, the
methodology, and the WRF-CHEM model description in Section 2. Results and discussions
were presented in Section 3, followed by the summaries and conclusions in Section 4.
**2 Data and methodology**
**2.1  The long-term NTL data and coal consumption**
In order to understand the spatial distributions of lighting before and after the GLP in China,
we investigated the version 4 of the Defense Meteorological Satellite Program Operational
Line Scanner (DMSP/OLS) NTL time series data from 1992 to 2013 (Elvidge et al., 2014).
The dataset available at: https://ngdc.noaa.gov/eog/dmsp/downloadV4composites.html. We
selected the stable light datasets, which are the cloud-free composites using all the archived
DMSP/OLS smooth resolution data for calendar years. The images represent the average
intensity of NTL with DN values ranging from 0 to 63 in 30 arc-second grids-cells (about 1 km
spatial resolution). The 1992 and the 2013 datasets were used to investigate the different
overview status of NTL before and after the GLP for long years. Considering the differences
between the sensors, differences in the crossing times of the satellites, and degradation of the
sensors (Elvidge et al., 2009; Elvidge et al., 2014), we inter-calibrated the NTL datasets





followed a second order regression model (Elvidge et al., 2014).
**Figure 1** shows the spatial distributions of the DMSP/OLS NTL data. We found that the
lighting usages were significantly increased from 1992 to 2013, both in lighting intensity and
spatial coverage, especially in the regions of eastern China, including the NCP, the Pearl River
Delta, and the Yangtze River Delta. The rapid increase in the usage of lighting suggested that
the generations of electricity were greatly enhanced.
**Figure 2** shows a long-term evolution of thermal power electricity and coal consumption for
power generation. It shows that the thermal power electricity increased from 2000 (about $10^{12}$
kW h) to 2015 (about $4\times10^{12}$ kW h), indicating that due to the rapid increase in the economics,
the demand of electricity was largely enhanced in China. The emission of $SO_2$ increased before
2006, due to the increase of coal consumption. While after 2006, although the coal
consumption still increased, the emission of $SO_2$ started to decrease, suggesting that the
desulfurization played important roles in the emission reductions from thermal power
generation (Liu et al., 2016). The decrease of NOx emission started to decrease in 2012, which
was 6 year later than the decrease in $SO_2$ emissions, suggesting that the denitrification played
important roles in the emission reduction from thermal power generation after 2012 (Hu et al.,
2016). Compared to the gas-phase emissions of $SO_2$ and NOx, the direct emission of particles
($PM_{2.5}$) was relatively small (Liu et al., 2015). The large portion of gas-phase emissions from
thermal power generation indicated that the most $PM_{2.5}$ emitted from the power plants might
be in the phase of secondary particles.
The above long-term variability of thermal power electricity and associated coal consumption
for power generation was based on the situation that the GLP was conducted in China, which
could produce a strong reduction for the coal burning emissions from thermal power



generation, such as air pollutants of $SO_2$ and NOx. These gases might have important effects
on the $PM_{2.5}$ pollution in China, because they are important precursors for the production of
particle matter (Seinfeld et al., 1998; Laurent et al., 2014). However, as the business as usual
condition (i.e., without the GLP), the increased lighting demand could cause significant
increase in thermal power electricity, and the associated growth of coal consumption for power
generation during the past decades. This study was to assess the potential effects induced by
the GLP on the severe haze pollution in the NCP (Tie et al., 2017; Long et al., 2018), and also
displayed a good example to illustrate that scientific innovation can induce important benefits
on environment issues. To assess the impacts of the GLP on the severe air polluted region in
China, such as in the NCP, several important tools and data were used in this study, including a
regional chemical/dynamical model (WRF-CHEM), satellite data (DMSP/OLS and S-NPP),
and surface measurements of air pollutants.
**2.2  Description of the WRF-CHEM model**
We used a specific version of the WRF-CHEM model (Grell et al., 2005). The model included
a new flexible gas-phase chemical module and the Models3 community multi-scale air quality
(CMAQ) aerosol module developed by the US EPA (Binkowski and Roselle, 2003). The
model included the dry deposition (Wesely 1989) and wet deposition followed the CMAQ
method. The impacts of aerosols and clouds on the photochemistry (Li et al., 2011b) were
considered by the photolysis rates calculation in the fast radiation transfer model (Tie et al.,
2003; Li et al., 2005). The inorganic aerosols (Nenes et al., 1998) were predicted using the
ISORROPIA Version 1.7. We also used a non-traditional secondary organic aerosol (SOA)
model, including the volatility basis-set modeling approach and SOA contributions from
glyoxal and methylglyoxal. Detailed information about the WRF-CHEM model can be found

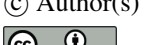



in previous studies (Li et al., 2010; Li et al., 2011a; Li et al., 2011b; Li et al., 2012).
In the present study, we simulated severe haze pollution from 1 to 31 December 2015 in the
NCP. The domain, centered at the point of (116° E, 38° N), was composed horizontally of 300
by 300 grid points spaced with a resolution of 6 km (**Fig. 3**) and vertically with 35 sigma levels.
The physical parameterizations included the microphysics scheme (Hong and Lim 2006), the
Mellor–Yamada–Janjic turbulent kinetic energy planetary boundary layer scheme (Janjić,
2002), the unified Noah land-surface model (Chen and Dudhia, 2001), the Goddard long wave
radiation parameterization(Chou and Suarez, 1999), and the shortwave radiation
parameterization (Chou et al., 2001). Meteorological initial and boundary conditions were
obtained from the 1° by 1° reanalysis data of National Centers for Environmental Prediction
(Kalnay et al., 1996). The spin-up time of WRF-CHEM model is 3 days. The chemical initial
and boundary conditions were constrained from the 6 h output of Model of Ozone and Related
chemical Tracers, Version 4 (Horowitz et al., 2003).
We utilized the anthropogenic emission inventory developed by Tsinghua University (Zhang et
al., 2009), including anthropogenic emission sources from transportation, agriculture, industry
and power generation and residential. The dataset can be accessible from the website of MEIC
(http://www.meicmodel.org), providing for the community a publically accessible emission
dataset over China with regular updates. The emission inventory used in the present study is
updated and improved for the year 2015. In addition, the emissions of $SO_2$, NOx, and CO have
been adjusted according to the observations during the period. Emissions from biogenic
sources were calculated online using the Model of Emissions of Gases and Aerosol from
Nature model (MEGAN) (Guenther et al. 2006).



### 2.3 Analysis of satellite data and model domain

Since the launch of the Suomi-National Polar-orbiting Partnership satellite in 2011, the

Day/Night Band for the Visible Infrared Imaging Radiometer Suite (VIIRS DNB) has been

widely used in recent studies, which confirmed to establish empirical relationships with energy

use (Román and Stokes, 2015; Coscieme et al., 2014). To some extent, the VIIRS NTL dataset

(in 15 arc-second grids-cells, about 500 m) are superior to the DMSP/OLS NTL dataset

(Elvidge et al., 2013). In the present study, we used the version 1 of VIIRS NTL dataset to

investigate the consumption of lighting electricity in each province, defined as provincial

dynamics as follow.

$$PD_i = \frac{\sum_i L_j \times S_j}{\sum_w L_j \times S_j} \tag{1}$$

where $i$ denotes the provincial domain, and $w$ is the nationwide domain. $j$ is the pixel of VIIRS

NTL dataset. $S$ is the area of pixel $j$. $L$ is the NTL radiance. The annual VIIRS NTL dataset

contains cloud-free average of NTL radiance by excluding any data impacted by stray light,

and further screening out the fires and other ephemeral lights and background (non-lights). The

dataset is available at: https://ngdc.noaa.gov/eog/viirs/download_dnb_composites.html.

The distribution of VIIRS NTL radiance in 2015 (**Fig. S1**) was similar as the DMSP/OLS DN

values (**Fig. 1**). The high values of annual NTL radiance were concentrated in the densely

populated and industrial developed areas of China (**Fig. S1a**), such as the NCP, the Yangtze

River Delta, and the Pearl River Delta. There were "hot spot" located in some megacities, such

as the Beijing, Tianjin, Shanghai, Guangzhou, where the NTL radiance can reach as high as 20

mW/m$^2$/sr. Statistically, 12.8% of these China's land areas consumes 58.3% of lighting

electricity consumption. The high values of provincial dynamics also concentrated on these

regions, and all the provincial dynamics exceeding 5% were coastal cities (**Fig. S1b**). In the



NCP, in addition to the high usage of lighting, there is a large amount of power plants (Liu et
al., 2015). We selected the NCP (**Fig. 3**) as the region of interest. In addition, there are
extensive measurement sites of pollutants in the domain (the green crosses in **Fig. 3**).
**2.4  Estimation of coal-saving induced by the GLP**
According to the analysis for the Chinese GLP program (Guo and Pachauri, 2017), the lighting
activities can be defined as three clusters according to their usages: **(C₁)** For outdoor lighting,
such as road lights; **(C₂)** household usage, mainly for residential applications; **(C₃)** commercial
and industrial buildings. In practice, the core of the GLP is to improve luminous efficiency,
replacing low-efficiency lighting lamps by high-efficiency ones. The details of the GLP
program were as follows. For **C₁**, the High Pressure Sodium lamps (HPS) and Metal Halide
(MH) lamps are primarily used to replace High Pressure Mercury-vapor lamps (HPM). For **C₂**,
the Compact Fluorescent Lamps (CFLs) are used to replace incandescent lamps (ILs). For **C₃**,
the T8/T5 fluorescent tubes are used to replace T12/T10 fluorescent tubes. The emerging LED
lamps were not covered, however, it promotes to each of the above cluster (Pan, 2018; Wang,
2017; Asolkar and Dr., 2017; Xie et al., 2016; Ge et al., 2016; Edirisinghe et al., 2016). Here
the LED lamps were allocated proportionally based on the proportions of the lighting
electricity consumption of **C₁**, **C₂**, and **C₃**.
According to the classification above, we estimated the current equivalent luminous efficacy
($ELE_{GLP}$) weighted by the proportion of their lighting electricity consumption. To investigate
the potential effectiveness of the GLP, we also calculated the equivalent luminous efficacy
without the implementation of the GLP ($ELE_{no-GLP}$).

246        $$ELE_{GLP} = \sum f_k LE_{k,GLP} \tag{2}$$

247        $$ELE_{no-GLP} = \sum f_k LE_{k,no-GLP} \tag{3}$$



where $k$ denotes the specified cluster of lighting lamps. $f_k$ is the proportion of lighting
electricity consumed by the $k^{th}$ cluster lamps; $LE_{k,GLP}$ and $LE_{k,no-GLP}$ denote the equivalent
luminous efficacy of the $k^{th}$ cluster lamps with and without the improvement of lighting
efficacy induced by the GLP. The ELE is a comprehensive parameter to reflect the lighting
efficacy. In terms of current consumption levels of lighting electricity, the lighting coal
consumption for power generation is proportional to ELE. As a result, the potential coal-saving
induced by the GLP ($dC$) can be estimated by:
$$dC = C_0 \times \frac{ELE_{no-GLP} - ELE_{GLP}}{ELE_{GLP}} \qquad (4)$$
where $C_0$ denotes the current coal consumption for thermal power generation. To get the
spatial distribution of potential provincial coal-savings ($dC_i$), we spatially scaled the total
potential coal-saving ($dC$) according to the provincial dynamics factor ($PD_i$), which is
calculated based on the spatiotemporal dynamic of electric power consumption in each
province (Elvidge et al., 1997; Chen and Nordhaus, 2011; He et al., 2013).
$$dC_i = dC \times PD_i \qquad (5)$$
where $i$ denotes the province; $PD_i$ reflects provincial dynamics of lighting coal consumption,
which was explained in **Eq. 1**.
To estimate the emission reduction induced by the GLP, we assumed that the potential
emission reduction was mainly due to the emissions from the thermal power plants. Based on
the current anthropogenic emission inventory from MEIC (Multi-resolution emission inventory
for China) (Liu et al., 2015; Zhang et al., 2009), the potential emission reduction ($dE_{power,spec}$)
induced by the GLP was proportional to the associated potential coal-saving for the thermal
power generation.
$$\frac{dE_{power,spec}}{dC} = \frac{E_{power,spec}}{C_0} \qquad (6)$$



where $E_{power,spec}$ denotes the emission inventory from the thermal power sector; *spec* is the
specify air pollutant of WRF-CHEM species. $dC$ and $C_0$ are the same as that in Eq. 4.
**2.5 WRF-CHEM sensitive studies**
Based on previous studies (Guo and Pachauri, 2017), the effective luminous efficacy (ELE)
increased from 50 lm/W to 70–140 lm/W for $\mathbf{C_1}$, from 15 lm/W to 50–60 lm/W for $\mathbf{C_2}$, and
from 70–80 lm/W to 80–105 lm/W for $\mathbf{C_3}$. Simultaneously, the LED has experienced a fast
growth since 2011, with the marketing share of LED lamps reached 32% in 2015, and the high
efficacy LED lamps with 150 lm/W had been industrialized production in China (Gao and
Zheng, 2016). Here we treated the marketing share of LED lamps as the proportion of its
lighting electricity consumption. Then it was allocated proportionally to the clusters according
to the research of Zheng et al., (2016), which reported the proportion of its lighting electricity
consumption with $\mathbf{C_1}$: $\mathbf{C_2}$: $\mathbf{C_3}$ being 31.6%: 19.7%: 48.7%. More detailed information can be
founded in **Table 1**.
The estimated ELE values have uncertainties for both low and high efficient lamps, ranging
from 52.8 to 57.7 lm/W and from 96.2 to 120.9 lm/W for the ELE with or without the GLP,
respectively (see **Table 1**). In addition, the estimate of lighting electricity accounts for 10–14%
of the total electricity (Zheng et al., 2016; Lv and Lv, 2012). As a result, the model sensitive
studies included low-limit and high-limit of electricity power savings. To account for all of the
uncertain ranges, in the lower limit model simulation, the thermal power was estimated to
increase 6.7%, without the GLP. In the higher limit model simulation, the thermal power was
estimated to increase 18.7%, without the GLP. **Figure 4** shows that under lower and higher
limit assumptions, the potential coal-savings induced by the GLP were 120–323 million tons,
respectively. According to these estimates into the reference emission inventory ($E_{0,spec}$), the





emission of pollutants, with the 3 cases (reference, low-limit, and high-limit) were estimated
and shown in **Table 2**. The reference emission inventory is developed by Tsinghua University
(Zhang et al., 2009), including current emission levels of thermal power plants (with
considering GLP).
**Table 2** also shows that the direct emission of $PM_{2.5}$ was much smaller than the direct
emission of $SO_2$ and NOx in gas-phase. The $PM_{2.5}$ concentrations included two different parts
from thermal power plants. One was from the direct emission of $PM_{2.5}$ in particle phase, and
the other was the secondary particle ($PM_{2.5}$), which was formed from the chemical
transformation from $SO_2$ and NOx. As a result, the large effect of the GLP on haze pollutions
was due to the changes in the emissions of $SO_2$ and NOx from the thermal power plants.
**3 Results and discussions**
**3.1  Model evaluation**
To better understand the effect of the GLP on the haze pollution in the NCP, we first
conducted an evaluation of the WRF-CHEM model performance. The modeled results were
compared to the hourly near-surface concentrations of CO, $SO_2$, $NO_2$, and $PM_{2.5}$. The data was
measured by the China's Ministry of Environmental Protection (MEP), and are accessible from
the website http://www.aqistudy.cn/. The locations of the measurement sites show in **Fig. 3**.
The model results were evaluated by calculating the following statistical parameters, including
normalized mean bias (*NMB*), the index of agreement (*IOA*), and the correlation coefficient (*r*).
These parameters were used to assess the performance of REF case in simulations against
measurements.
$$NMB = \frac{\sum_{i=1}^{N}(P_i - O_i)}{\sum_{i=1}^{N} O_i} \qquad\qquad (7)$$





$$IOA = 1 - \frac{\sum_{i=1}^{N}(P_i - O_i)^2}{\sum_{i=1}^{N}(|P_i - \bar{O}| + |O_i - \bar{O}|)^2}$$
(8)

$$r = \frac{\sum_{i=1}^{N}(P_i - \bar{P})(O_i - \bar{O})}{[\sum_{i=1}^{N}(P_i - \bar{P})^2 \sum_{i=1}^{N}(O_i - \bar{O})^2]^{\frac{1}{2}}}$$
(9)

where $P_i$ and $O_i$ are the calculated and observed air pollutant concentrations respectively. $N$
is the total number of the predictions used for comparisons. $\bar{P}$ and $\bar{O}$ represent the average
predictions and observations, respectively. The *IOA* ranges from 0 to 1, with 1 showing perfect
agreement of the prediction with the observation. The *r* ranges from -1 to 1, with 1 implicating
perfect spatial consistency of observation and prediction.
**Figure 5** shows the temporal variation of modeled results with the measured values during
December 2015. The measured values of pollutants ($PM_{2.5}$, $NO_2$, $SO_2$, and CO) averaged in the
NCP were compared with the modeled results. The results indicate that there were strong
episodes of the hourly $PM_{2.5}$ mass  concentrations, with the highest values of exceeding 300 µg
$m^{-3}$, implicating that several haze events occurred during the period. There are several peak
values of $PM_{2.5}$ concentrations occurred during period, with a highest peak occurred between
22-24[th] December. Comparing with CO temporal variability, the temporal variations between
CO and $PM_{2.5}$ were similar. The modeled $PM_{2.5}$ and CO captured the strong temporal variation,
with the *IOA* of 0.98 and the *NMB* of 1.3% for $PM_{2.5}$ mass  concentrations and *IOA* of 0.89 and
the *NMB* of 4.3% for CO mass  concentrations. Since the CO variability was mainly
determined by meteorological conditions, the similarity of the temporal variability suggested
that the meteorological conditions had important contribution to the several peak values of the
episode, and the model simulation well captured the meteorological conditions during the
study period.
Although there was a similarity of the temporal variability between $PM_{2.5}$ and CO, the
magnitude of the variability of CO was smaller than variability of $PM_{2.5}$, suggesting that in



addition to the meteorological conditions, the chemical formation also played important roles
for producing the high peaks of $PM_{2.5}$ concentrations. It is important to simulate the measured
temporal variations of $SO_2$ and NOx, because they are important chemical precursors (Seinfeld
and Pandis, 1998; Laurent. et al., 2014), and are the major pollutants emitted from the thermal
power plants (**Table 2**). As shown in **Fig. 5**, both the measured and modeled $SO_2$ and NOx had
several episodes, which were corresponding to the episodes of the $PM_{2.5}$. The parameters
between the measured and modeled results were acceptable, with the *IOA* of 0.83 and the *NMB*
of 1.3% for $SO_2$, and *IOA* of 0.93 and the *NMB* of 6.1% for NOx. It is interesting to note that
the occurrences of the peak of $SO_2$ and NOx are about 1-2 days ahead of the peak of $PM_{2.5}$.
One of the explanations was that there was chemical conversion from gas-phase of $SO_2$ and
NOx to particle phase of $PM_{2.5}$, resulting in the time lag between the peaks of $SO_2$-NOx and
$PM_{2.5}$, because $SO_2$ and NOx were the precursors of $PM_{2.5}$ (Seinfeld and Pandis, 1998; Laurent.
et al., 2014). As we state in the previous sections, the large effect of the GLP on haze
pollutions was due to the changes in the emissions of $SO_2$ and NOx from the thermal power
plants. The good statistical performance of the modeled $SO_2$ and NOx provided confident to
use the model to study the GLP effects on haze in the NCP region.
In order to do more thoughtful validation of the model performance, **Figure 6** shows the
measured and modeled spatial distributions of $PM_{2.5}$, $SO_2$, and NOx in the NCP. The model
generally reproduced the spatial variations of $PM_{2.5}$, $NO_2$, and $SO_2$, capturing the spatial
characters. For example, the $SO_2$ were largely emitted from thermal power plants and steel
industrials, which were large point sources. As a result, both the modeled and measured $SO_2$
appeared as scattered distributions (see **Fig. 6d**). The correlation coefficients (*r*) between the
measured and modeled results were 0.86, 0.68, and 0.70 for $PM_{2.5}$, $NO_2$, and $SO_2$, respectively.



In general, the NCP encountered severe haze pollution events during the December 2015. The
statistical analysis showed that the WRF-CHEM model reasonably captured the spatial and
temporal variations of haze pollution in the NCP, although some model biases existed. The
model validation provided a confident to the further model studies.
**3.2  Potential benefit of the GLP to air pollution in the NCP**
There are massive emissions of NOx and $SO_2$ from thermal power plants in the research
domain, producing 299.1 Gg and 103.7 Gg (**Tab. 1**) during the December 2015, for NOx and
$SO_2$, respectively. There is more emission amount of NOx than $SO_2$, because the $SO_2$
emissions from power had been significantly declined since 2005, whereas the NOx emissions
were slightly declined (see **Fig. 2**) due to lower effective NOx emission control facilities (Liu
et al., 2015; Huang et al., 2016).
According to the estimate of 6.7–18.0% of potential coal-saving induced by the GLP (**Sect.**
**2.5**), the potential emission reductions from power generation were calculated base on **Eq. 6**,
and the emission reductions of NOx and $SO_2$ induced by the GLP were estimated for the
WRF-CHEM model sensitive studies. **Figure 7** shows the spatial distributions of changes in
NOx and $SO_2$ emissions in the research domain, especially the provinces of Hebei, Henan, and
Shandong within the NCP, where concentrated most of the power plants (Liu et al., 2015). The
results show that under low limit estimate, without the GLP, the NOx and $SO_2$ emissions
would be increased by 20.0 Gg and 6.9 Gg, respectively, in December 2015. Under high limit
estimate, without the GLP, the NOx and $SO_2$ emissions would be increased by 53.8 Gg and
18.7 Gg in the NCP. These large emission changes without the GLP could cause important
effects on the air pollution. In the following sections, the GLP effect on the reduction of air
pollution was investigated by using the WRF-CHEM model.

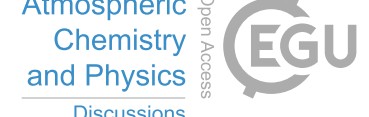



According to the lower and upper limits of emission reductions induced by the GLP, we
evaluated their resultant effects on air pollutants ($PM_{2.5}$, $NO_2$, and $SO_2$), which are estimated
by the difference of the SEN-GLP cases and the REF case (**Fig. 8**). The result shows that the
GLP has important effects on $PM_{2.5}$ concentrations (see **Figs 8a and 8b**), implicating the
remarkable benefit to haze pollution in the NCP. In the lower limit case, the $PM_{2.5}$
concentrations could be decreased by 2–5 µg m$^{-3}$ in large areas within the NCP, such as the
southeastern Hebei, northeastern Henan, and western Shandong (**Fig. 8a**). In the upper limit
case, there is much more remarkable decrease in $PM_{2.5}$ concentrations (4–10 µg m$^{-3}$) in wider
areas within the NCP (**Fig. 8b**). We can also find large-scale reductions of $NO_2$ and $SO_2$ in the
NCP (**Fig. 8c-f**). For example, in high limit case, the reduction of $NO_2$ ranges from 1–8 µg m$^{-3}$,
and the reduction of $SO_2$ ranges from 1–4 µg m$^{-3}$. We also display the species variations ($PM_{2.5}$,
$NO_2$, and $SO_2$) in **Fig. S2** within the areas with high $PM_{2.5}$ changes induced by the GLP (see
red-square in Fig. 8).
Although the influence of the GLP is to decrease $PM_{2.5}$ concentrations, there were some slight
increase in $PM_{2.5}$ concentrations in north of NCP. As indicated in **Table 2**, the directly
emission of $PM_{2.5}$ was less than the gas-phase emissions of NOx and $SO_2$, which suggested that
the decrease of $PM_{2.5}$ by applying the GLP was mainly due to the chemical conversions from
gas-phase NOx and $SO_2$ to nitrate and sulfate particles (Seinfeld et al., 1998; Laurent et al.,
2014). The slight increase of the $PM_{2.5}$ concentrations may be induced by the changes in $O_3$
concentrations, because the chemical conversion from NOx and $SO_2$ to nitrate and sulfate
requires the atmospheric oxidants like $O_3$. As shown in **Fig. S3**, there is slight increase of $O_3$
(1–2 µg m$^{-3}$) due to the GLP, and the slightly increase the oxidation of $SO_2$, which may cause
some enhancement of sulfate concentrations (Wang et al., 2015a; Xue et al., 2016). Apparently,

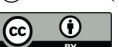



the NO$_2$ reductions are more remarkable because of the more noteworthy NOx emission
reductions induced by the GLP.
The GLP resulted in significant reduction of potential pollutant emissions from the thermal
power generation, corresponding to potential benefit in alleviating haze pollution in the NCP,
although with few fluctuated deteriorations. It also benefits the pollution of NOx and SO$_2$ in
the NCP.
**4 Summary**
For replacing low-efficiency lighting lamps by high-efficiency ones, the Green Lights Program
(GLP) is a national energy conservation activity for saving lighting electricity consumption in
China, resulting in an effective reduction of coal consumption for power generation. However,
despite of the great success of the GLP in lighting electricity, the effects of the GLP on haze
pollution are not investigated and well understood. In the present study, we try to assess the
potential coal-saving induced by the GLP, and to estimate its resultant benefit to the haze
pollutions in the NCP, China, where often suffer from severe haze pollutions. First, we used
the satellite dataset of nighttime lights to evaluate the associated saving of lighting electricity
consumption and its resultant coal-saving in the NCP. Second, we estimated the emission
reductions from thermal power generation induced by the GLP, based on the emission
inventory developed by Tsinghua University (Zhang et al., 2009). Finally, we applied the
WRF-CHEM model to evaluate the potential effects of the GLP on the haze pollutions in the
NCP. The model results had been evaluated by a comparison with surface measurements. And
two sensitivity experiments were conducted to explore the role of the GLP in benefiting the
haze pollution. Some important results are summarized as follows.
1. Due to the rapid increase in the economics, the demand of electricity is largely enhanced in




China. As a result, the thermal power electricity increase from 2000 (about $10^{12}$ kW h) to
2015 (about $4\times10^{12}$ kW h), suggesting that the lighting electricity consumption could
produce higher emissions of air pollutions in the densely populated and industrial developed
regions of China.
2. The GLP program significantly improves in lighting efficiency by 66.8–128.8%,
implicating 6.7–18.0% of potential savings for electricity consumption, as well as potential
coal-savings in thermal power generation.
3. The estimated potential coal-saving induced by the GLP can reach a massive value of 120–
323 million tons, accounting for 6.7–18.0% of the total coal consumption for thermal power
generation in China. As a result, there is a massive potential emission reduction of air
pollutants from thermal power generation, involving 20.0–53.8 Gg for NOx and 6.9–18.7
Gg for $SO_2$ in the NCP of China. The reductions of these emissions play important roles in
reducing the haze formation in the NCP, because NOx and $SO_2$ are important precursors for
the particles.
4. The reduction of NOx and $SO_2$ from power plants produces a remarkable benefit to haze
pollution in the NCP. The sensitive studies by using the WRF-CHEM model shows that the
GLP has important effects on $PM_{2.5}$ concentrations in the NCP. In the lower limit case, the
$PM_{2.5}$ concentrations could be decreased by 2–5 $\mu g\ m^{-3}$ in large areas within the NCP. In the
upper limit case, there is much more remarkable decrease in $PM_{2.5}$ concentrations (4–10 $\mu g$
$m^{-3}$) in wider areas within the NCP.
This study is a good example to illustrate that scientific innovation can induce important
benefits on environment issues, such as haze pollution.





## Author contributions


X. T., and X. L. designed the study. X.-K. L. provided measurement data. J.-M.Z., W.-T. D.,
F. T., G.-H. L. analyzed the data. X. L. and X. T. wrote the manuscript. J. C. and Z. A.
overviewed the paper. All authors commented on the manuscript.

## Acknowledgement


This work is supported by the National Natural Science Foundation of China (NSFC) under
Grant No. 41430424 and 41730108, and the Ministry of Science and Technology of China
under Grant No. 2016YFC0203400. The National Center for Atmospheric Research is
sponsored by the National Science Foundation.

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




**Tab.1**

618                **Table 1.** Effective Luminous Efficacy (ELE) with and without the GLP

| | Cluster[a] | Lamp type | LE[a] | P[b] | ELE |
|---|---|---|---|---|---|
| Low-efficiency lamps | | | | | $ELE_{no-GLP}$ |
| Lower range | C1 | HPM | 50[a] | 31.6%[b] | 52.8 |
| | C2 | ILs | 15[a] | 19.7%[b] | |
| | C3 | T12/T10 | 70 | 48.7%[b] | |
| Upper range | C1 | HPM | 50[a] | 31.6%[b] | 57.7 |
| | C2 | ILs | 15[a] | 19.7%[b] | |
| | C3 | T12/T10 | 80[a] | 48.7%[b] | |
| High-efficiency lamps | | | | | $ELE_{GLP}$ |
| Lower range | C1, C2, C3 | LED | 150[c] | 32.0%[c] | 96.2 |
| | C1 | HPS/MH | 70[a] | 21.5%[d] | |
| | C2 | CFLs | 50[a] | 13.4%[e] | |
| | C3 | T8/T5 | 80[a] | 33.1%[e] | |
| Upper range | C1, C2, C3 | LED | 150[c] | 32.0%[c] | 120.9 |
| | C1 | HPS/MH | 140[a] | 21.5%[d] | |
| | C2 | CFLs | 60[a] | 13.4%[e] | |
| | C3 | T8/T5 | 105[a] | 33.1%[e] | |

P: the proportion of lighting electricity consumed by specific cluster lamps to the total lighting electricity
consumption
a. The values were taken from Guo et al. (2017).
b. The values were taken from Zheng et al. 2016
c. The values were evaluated based on Gao et al., 2016
d. The values were estimated based on Zheng et al., 2016 and Ding et al., 2017
e. The values were estimated based on Refs of a, b, c, and d.
LE and ELE: (lm/W)
LED: light-emitting diode
HPM lamps: High Pressure Mercury-vapor lamps
HPS lamps: High Pressure Sodium lamps          MH lamps: Metal Halide
ILs: Incandescent lamps                        CFLs: Compact Fluorescent Lamps
T12/T10: T12/T10 fluorescent tubes             T5/T8: T5/T8 fluorescent tubes
C1: outdoor lighting, such as road lights
C2: residential applications, such as households
C3: commercial and industrial buildings



**Tab.2**

**Table 2.** Coal consumptions, and emissions for the reference case (REF), the limit cases of low

641                (SEN-GLP-low) and high (SEN-GLP-high)


| Species | REF | SEN-GLP-low | SEN-GLP-high |
|---|---|---|---|
| | (100%) | (+6.7%) | (+18.0%) |
| Coal consumption for coal-fired power in China in 2015 (Tg) | | | |
| | 1793.2 | 119.7 | 323.3 |
| Emissions from power in 3 cases in the domain in Dec. 2015 (Gg) | | | |
| NOx | 299.1 | 299.1+20.0 | 299.1+53.8 |
| $SO_2$ | 103.7 | 103.7+6.9 | 103.7+18.7 |
| $PM_{2.5}$ | 31.1 | 31.1+2.1 | 31.1+5.6 |
| Others | X | 106.7X% | 118.0X% |






**Figure Captions**
**Figure 1**. The spatial distributions of the Nighttime-light data (NLT) from DMSP/OLS DN
values in (a) 1992 and in (b) 2013.
**Figure 2**. Coal-fired power electricity and associated coal consumption for power generation,
and the emissions of NOx, $SO_2$, and $PM_{2.5}$ from thermal power plants from 2000 to
2015 in China.
**Figure 3**. The horizontal domain of the model (WRF-CHEM), with the location of sampling
sites (shown by the green crosses), and topographical conditions of the NCP, which
are surrounded by the Mountains of Yan and Tai in the north and west, respectively.
**Figure 4**. The **(a)** lower and **(b)** upper limits of potential coal-savings induced by the GLP.
**Figure 5**. The temporal variations of predicted (red lines) and observed (black dots) profiles of
near-surface mass concentrations of $PM_{2.5}$, $NO_2$, $SO_2$, and CO averaged over all
ambient monitoring sites in the NCP during December 2015.
**Figure 6.** The spatial comparisons of predicted and observed episode-average mass
concentrations of $PM_{2.5}$, $NO_2$, and $SO_2$. **(a)** Statistical comparison of predicted and
observed mass concentrations, with the correlation coefficient (*r*). Horizontal
distributions of predictions (color contour) and observations (colored circles) of **(b)**
$PM_{2.5}$, **(c)** $NO_2$, and **(d)** $SO_2$, along with the simulated wind fields (black arrows).
**Figure 7.** The potential emission reductions for low (left panels) and high (right panels) limit
cases induced by the GLP, including the mass rates change of **(a)** $NO_x$, and **(b)** $SO_2$.
The total emission reductions are also shown in the rectangle.
**Figure 8.** The lower (left panels) and upper (right panels) episode-averaged variations induced
by GLP, including the mass concentrations ($\mu g\ m^{-3}$) of **(a)** $PM_{2.5}$, **(b)** $NO_2$, and **(c)**
$SO_2$. The results refer to the spatial variations between the REF case and the
SEN-GLPs case (REF − SNE-GLPs).



**Fig. 1**

**Figure 1**. The spatial distributions of the Nighttime-light data (NLT) from DMSP/OLS DN
values in (a) 1992 and in (b) 2013.





**Fig. 2**

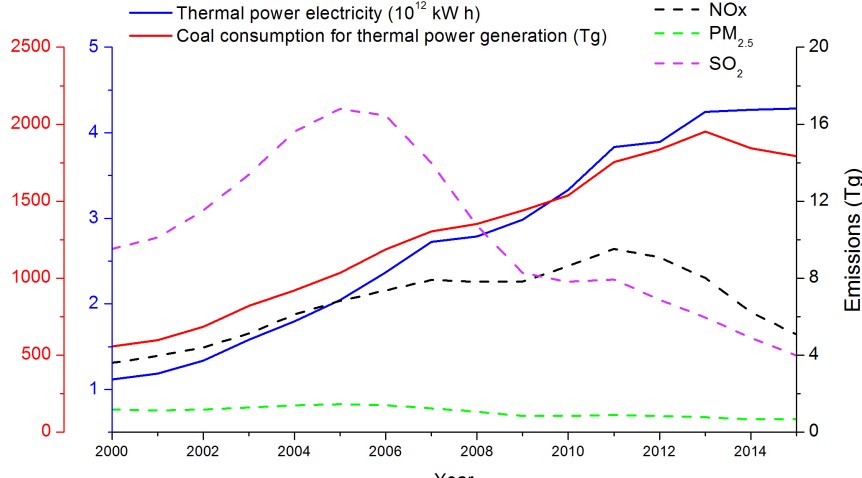


**Figure 2**. Coal-fired power electricity and associated coal consumption for power generation,
and the emissions of NOx, $SO_2$, and $PM_{2.5}$ from thermal power plants from 2000 to 2015 in
China.




**Fig. 3**

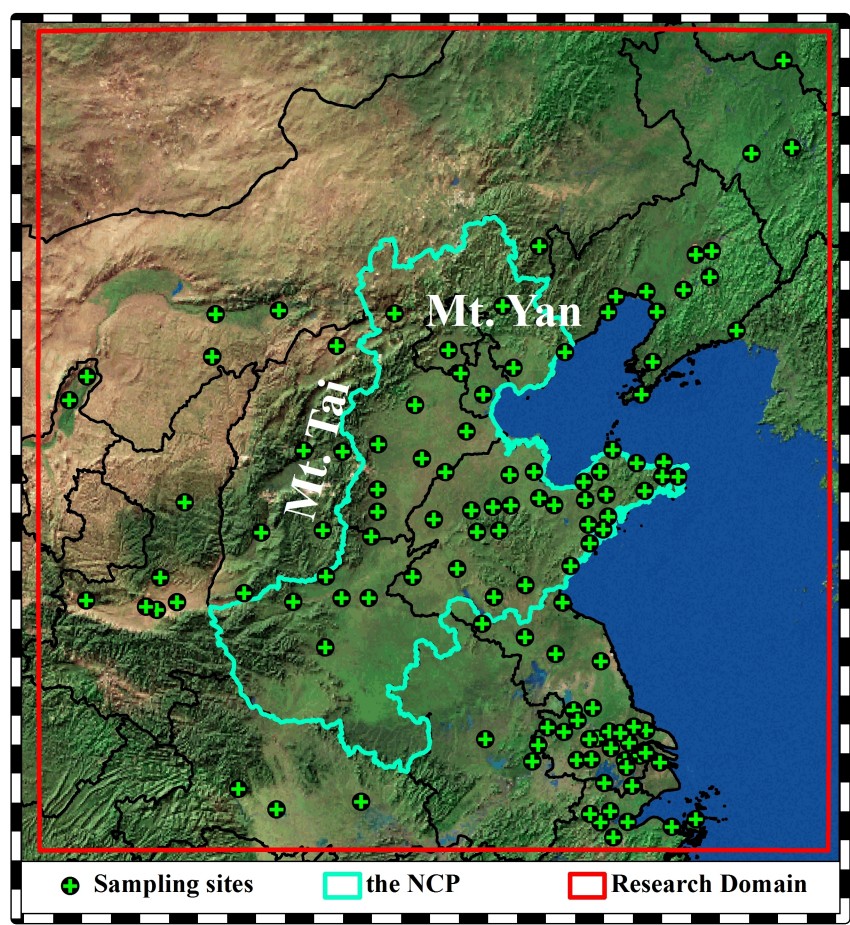


**Figure 3**. The horizontal domain of the model (WRF-CHEM), with the location of sampling
sites (shown by the green crosses), and topographical conditions of the NCP, which are
surrounded by the Mountains of Yan and Tai in the north and west, respectively.



**Fig. 4**



**Figure 4**. The **(a)** lower and **(b)** upper limits of potential coal-savings induced by the GLP.





**Fig. 5**

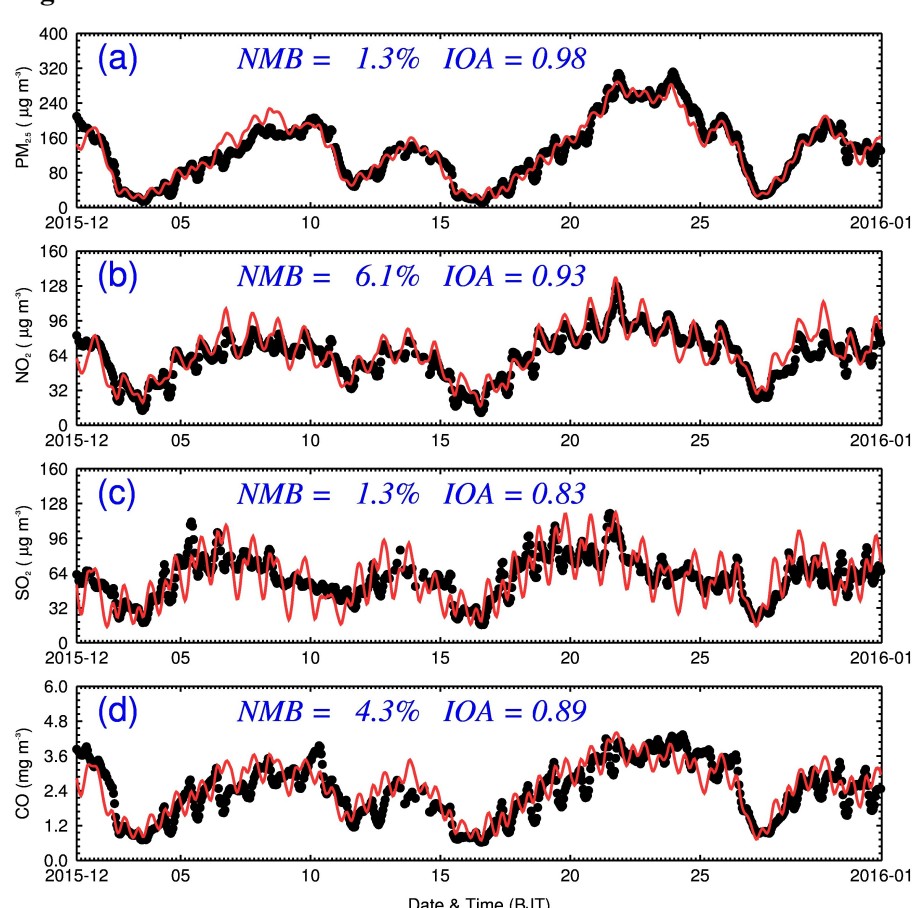


**Figure 5**. The temporal variations of predicted (red lines) and observed (black dots) profiles of
near-surface mass concentrations of $PM_{2.5}$, $NO_2$, $SO_2$, and CO averaged over all ambient
monitoring sites in the NCP during December 2015.





**Fig. 6**

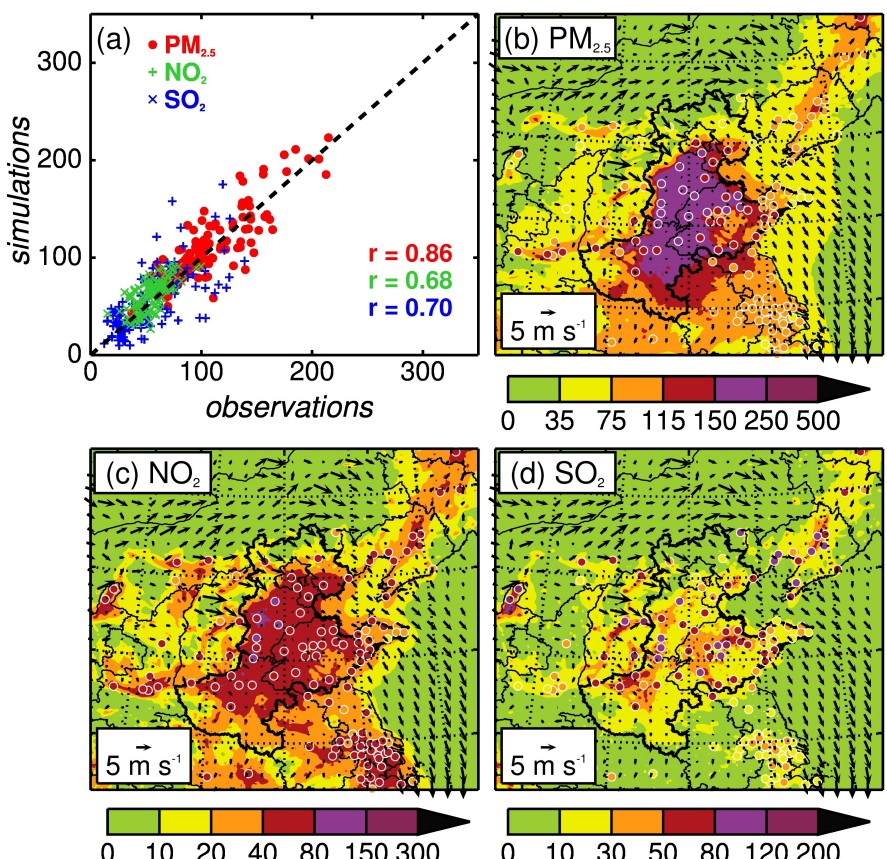


**Figure 6.** The spatial comparisons of predicted and observed episode-average mass
concentrations of $PM_{2.5}$, $NO_2$, and $SO_2$. **(a)** Statistical comparison of predicted and observed
mass concentrations, with the correlation coefficient (*r*). Horizontal distributions of predictions
(color contour) and observations (colored circles) of **(b)** $PM_{2.5}$, **(c)** $NO_2$, and **(d)** $SO_2$, along
with the simulated wind fields (black arrows).





**Fig. 7**

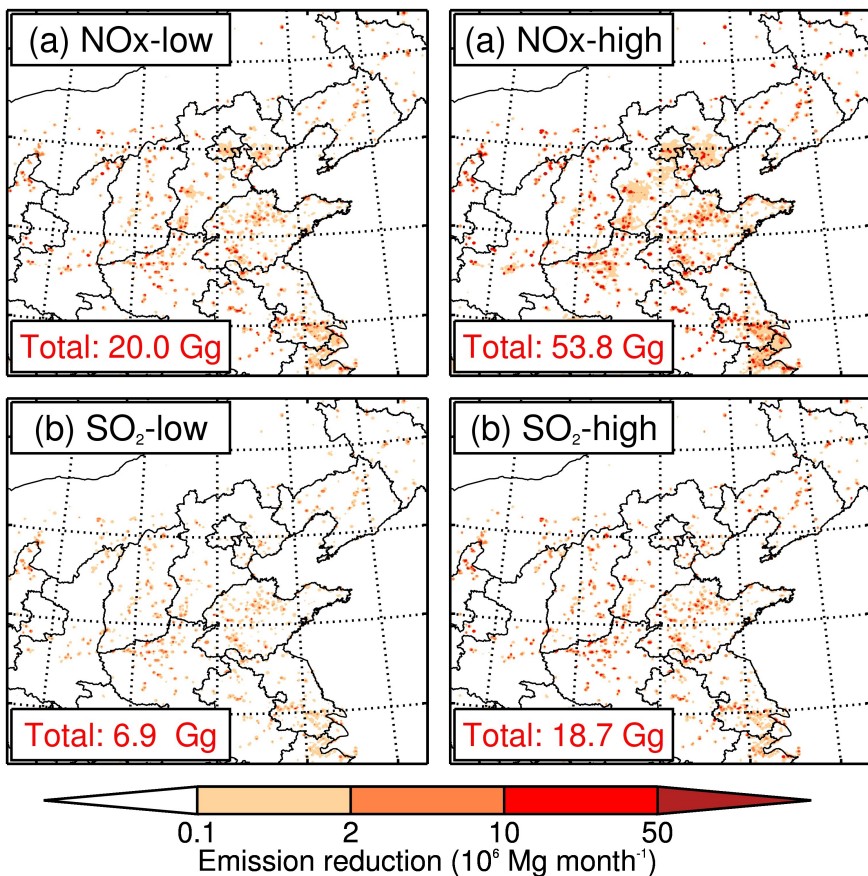


**Figure 7.** The potential emission reductions for low (left panels) and high (right panels) limit
cases induced by the GLP, including the mass rates change of **(a)** $NO_x$, and **(b)** $SO_2$. The total
emission reductions are also shown in the rectangle.





**Fig. 8**

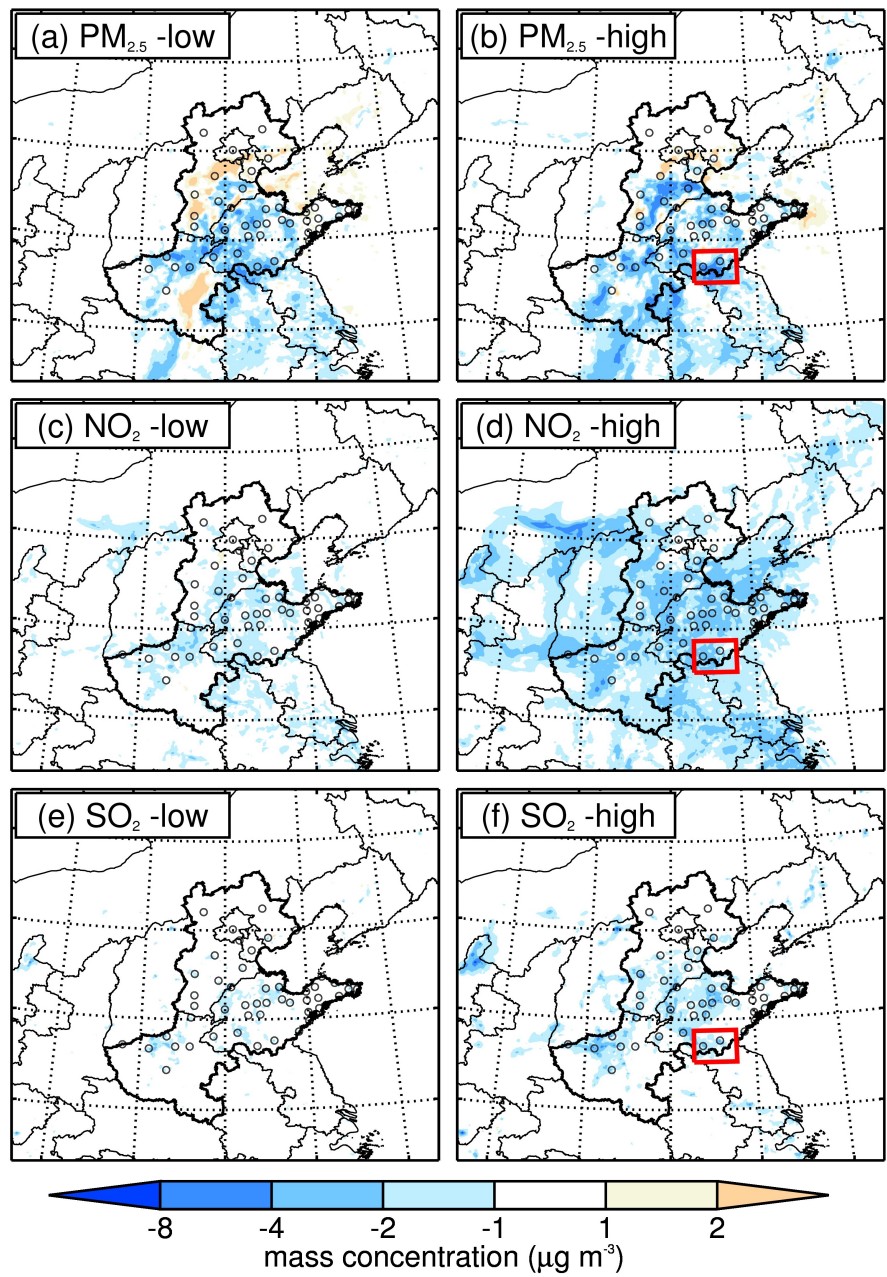


**Figure 8.** The lower (left panels) and upper (right panels) episode-averaged variations induced

by GLP, including the mass concentrations ($\mu g\ m^{-3}$) of **(a)** $PM_{2.5}$, **(b)** $NO_2$, and **(c)** $SO_2$. The

results refer to the spatial variations between the REF case and the SEN-GLPs case (REF −

SNE-GLPs). The red-squares display the areas with high $PM_{2.5}$ changes induced by the GLP.