# Peer review of "Impact of the Green Light Program on haze in the North"

_Atmospheric Chemistry and Physics, 2018_

## Referee Comment (RC1) · Anonymous Referee #2 · 21 May 2019

Green Light Program in China on air quality was not evaluated. The subject of this study is valuable and has potential value on air pollution control. Some minor suggestions that the authors may consider to follow. 1) Haze is one kind of phenomenon in the meteorological record. Normally we say haze, hazy day, but not reasonable to say haze pollution. I suggest that the paper instead haze pollution of haze or aerosol/PM2.5 pollution. 2) In line 147-148, what's the reason of "The decrease of NOx emission was 6 year later than the decrease in SO2 emissions"? Although, denitrification in thermal power generation after 2012 (Hu et al., 2016), the NOx emission from transportation was increase much in this period. 3) In the thermal power generation, there should be large differences in the air pollutants treatment technology in 2001 and 2010. I wonder whether the study consider the coal-saving induced by the GLP in the condition of different purification efficiency of air pollutants (SO2 and NOx etc.) in thermal power generation in 2001 and 2010. 4) In the introduction, I suggest give a brief review of emission reduction of air pollutants on the structure of boundary layer and its impact on other species, eg. O3. Two of the references related: Li Z., et. al, Aerosol and boundary-layer interactions and impact on air quality, National Science Review, 4, 810-833, doi:10.1093/nsr/nwx117, 2017. Gao J., et al. "Effects of black carbon and boundary layer interaction on surface ozone in Nanjing, China." Atmospheric Chemistry and Physics 18.10(2018):7081-7094.

---

## Referee Comment (RC2) · Anonymous Referee #1 · 17 Jun 2019

This paper assessed the air quality impact of the Green Lights Program (GLP) in China using WRF-CHEM. The topic is interesting and has not been fully investigated by previous studies. However, I have big concern about the reliability of the results, due to the large uncertainties of the assumptions and methodology adopted by the study. I also have the impression that the study focused on the air quality impact of power plants, which has been explored by many existing work, but not GLP. I would recommend substantial improvements to infer the impact of GLP, but not power plants.

General comments:

1. Introduction: The authors tried to emphasis the importance of lights based on the fact that power plants play significant role in air pollution. However, what the ratio of power electricity goes to lights? This information is not clear to me after reading the

introduction. The same problem for the contents on Page 7, line 138-139.

2. Section 2.1. This section needs to be re-organized. The authors claim conclusions without showing data support. For instance, Page 7, line 154 and afterwards. "The above long-term variability of thermal power electricity and associated coal consumption for power generation was based on the situation that the GLP was conducted in China." The impact of GLP on power plants has not been quantified when stating so.

3. Page 12, line 264. "To estimate the emission reduction induced by the GLP, we assumed that the potential emission reduction was mainly due to the emissions from the thermal power plants." The uncertainty of this assumption is missing.

4. Regional diversity of LED market share. The emission rates of power plants have large diversities over regions in China. If the LED market share has large variations over regions as well, the derived emission reductions will be significantly different from what was estimated in the current study.

5. Secondary particles in Table. The method of calculating secondary particles from power plants is not clear to me.

Specific comments:

1. abstract. "In the upper limit case of emission reduction..." and "in the lower limit case of emission reduction..." The English sounds not correct for me.

2. line 77-78, please consider rephrasing "the famous nation-wide project of utilizing emission control facilities" . famous?

3. line 88. Missing a space before 2005.

4. line 92-93. I don't quite get the meaning of the sentence. Please consider rephrasing it.

5. line 105. The English is incorrect.

---

## Author Comment (AC1) · 22 Jul 2019

**Response to Referee #2**

We thank the reviewers for the careful reading of the manuscript and helpful comments. According to the suggestions of the reviewer, the reviewers' comments have been carefully addressed, and the paper is carefully revised. We believe that the revised paper has been significantly improved after addressing the comments of the reviewers.

**Comment:** *Green Light Program in China on air quality was not evaluated. The subject of this study is valuable and has potential value on air pollution control. Some minor suggestions that the authors may consider to follow. 1) Haze is one kind of phenomenon in the meteorological record. Normally we say haze, hazy day, but not reasonable to say haze pollution. I suggest that the paper instead haze pollution of haze or aerosol/PM$_{2.5}$ pollution.*

**Response**: Thanks for the valuable comments, and we replaced "haze pollution" with "haze" in the text.

**Comment:** 2) *In line 147-148, what's the reason of "The decrease of NO$_x$ emission was 6 year later than the decrease in SO$_2$ emissions"? Although, denitrification in thermal power generation after 2012 (Hu et al., 2016), the NO$_x$ emission from transportation was increase much in this period.*

**Response**: We agreed the reviewer that the NO$_x$ emission from transportation was still increase after 2012. However, **Figure 2** do not include NOx emissions from transport, only showing the NO$_x$ emissions from thermal power plants, and the data respected to the previous studies of *Liu et al., 2015 (Fig. 4)* and *Tong et al., 2018 (Fig. 2A)*. We revised inexact description to explain more clearly **in Line-154:** "Distinguished from the increase trend of NOx emission from transportation (Hu et al., 2016), the decrease of NO$_x$ emission from power sector started to decrease in 2012

due to the significant technological improvement of coal-consumption weighted mean NOx removal efficiency (Hu et al., 2016; Tong et al., 2018)."

**Comment:** *3) In the thermal power generation, there should be large differences in the air pollutants treatment technology in 2001 and 2010. I wonder whether the study consider the coal-saving induced by the GLP in the condition of different purification efficiency of air pollutants ($SO_2$ and $NO_x$ etc.) in thermal power generation in 2001 and 2010.*

**Response**: We agreed with the reviewer that the there should be large differences in the air pollutants treatment technology in the thermal power generation. Between the base case (with the GLP) and sensitivity cases (without the GLP), we focus on the potential emission reductions derived by the potential lighting electricity savings induced by the GLP, excluding other influence factors, and estimated with the same purification efficiency of air pollutant emission. We added descriptions of this issue **in Line-268:** "It is worth noting that, in the present study, we focused on the potential emission reductions derived by the potential lighting electricity savings induced by the GLP. And the emission reduction was confined at the improvement of luminous efficacy, which is the core of the GLP (Guo et al., 2017). Between the base case (with the GLP) and sensitivity cases (without the GLP), the coal-saving induced by the GLP was estimated with the same purification efficiency of air pollutant emissions between the base case (with the GLP) and sensitivity cases (without the GLP). And the ratio of power electricity goes to lights is same with the ratio of artificial lighting to the total electricity consumption, which is 10–14% (Lv and Lv, 2012; Zheng et al., 2016)."

**Comment:** *4) In the introduction, I suggest give a brief review of emission reduction of air pollutants on the structure of boundary layer and its impact on other species, eg. $O_3$. Two of the references related: Li Z., et. al, Aerosol and boundary-layer interactions and impact on air quality, National Science Review, 4, 810-833, doi:10.1093/nsr/nwx117, 2017. Gao J., et al. "Effects of black carbon and boundary layer interaction on surface ozone in Nanjing, China." Atmospheric Chem- istry and Physics 18.10(2018):7081-7094.*

**Response**: We added a brief review of emission reduction of air pollutants on the structure of boundary layer and its impact on other species and add some reference regarding the discussion **in Line-82:** "Emission reductions of air pollutants can substantially reduce the aerosol loading, and thus influenced the boundary layer, which is inherently connected to air pollution (Li et al., 2017). The interactions between aerosol and boundary layer can influence the surface ozone significantly, and more attention should be paid when controlling ozone pollution (Gao et al., 2018)."

**References**

Tong, D., Zhang, Q., Liu, F., Geng, G., Zheng, Y., Xue, T., Hong, C., Wu, R., Qin, Y., Zhao, H., Yang, L., He, K., 2018. Current Emissions and Future Mitigation Pathways of Coal-Fired Power Plants in China from 2010 to 2030. Environmental Science & Technology 52, 12905-12914.

Li, Z., Guo, J., Ding, A., Liao, H., Liu, J., Sun, Y., ... & Zhu, B., 2017. Aerosol and boundary-layer interactions and impact on air quality. National Science Review, 4(6), 810-833.

Gao, J., Zhu, B., Xiao, H., Kang, H., Pan, C., Wang, D., & Wang, H. 2018. Effects of black carbon and boundary layer interaction on surface ozone in Nanjing, China. Atmospheric Chemistry and Physics, 18(10), 7081-7094.

---

## Author Comment (AC2) · 22 Jul 2019

**Response to Referee #1**

We thank the reviewers for the careful reading of the manuscript and helpful comments. According to the suggestions of the reviewer, the reviewers' comments have been carefully addressed, and the paper is carefully revised. We believe that the revised paper has been significantly improved after addressing the comments of the reviewers.

\*\*\*\*\*\*\*\*\*\*\*\*\*\*\*\*\*\*\*\*\*\*\*\*\*\*\*\*\*\*\*\*\*\*\*\*\*\*\*\*\*\*\*\*\*\*\*\*\*\*\*\*\*\*\*\*\*\*\*\*\*\*\*\*\*\*\*\*\*\*\*\*\*\*\*

**Comment:** *This paper assessed the air quality impact of the Green Lights Program (GLP) in China using WRF-CHEM. The topic is interesting and has not been fully investigated by previous studies. However, I have big concern about the reliability of the results, due to the large uncertainties of the assumptions and methodology adopted by the study. I also have the impression that the study focused on the air quality impact of power plants, which has been explored by many existing work, but not GLP. I would recommend substantial improvements to infer the impact of GLP, but not power plants.*

**Response**: Thanks for the comments. In the revised manuscript, we focus on the effect on GLP, rather than power plants. More details mentioned by the reviewer are added in the revised version.

(1) We added the logical explanations between the GLP and the air quality improvement **in Line-162:** "The GLP focused on improving the luminous efficacy, saving lighting electricity, and thus reducing the coal consumption and air pollutant emissions from thermal power generation, which is inherently connected to air quality."

(2) We added more description of method **in Line-268:** "Here we focus on the potential emission reductions derived from the potential lighting electricity savings induced by the GLP. And the emission reduction was confined at the improvement of luminous efficacy, which is the core of the GLP (Guo et al., 2017). Between the base case (with the GLP) and sensitivity cases (without the GLP), the coal-saving induced

by the GLP was estimated with the same purification efficiency of air pollutant emissions between the base case (with the GLP) and sensitivity cases (without the GLP). And the ratio of power electricity goes to lights is same with the ratio of artificial lighting to the total electricity consumption, which is 10–14% (Lv and Lv, 2012; Zheng et al., 2016)."

\*\*\*\*\*\*\*\*\*\*\*\*\*\*\*\*\*\*\*\*\*\*\*\*\*\*\*\*\*\*\*\*\*\*\*\*\*\*\*\*\*\*\*\*\*\*\*\*\*\*\*\*\*\*\*\*\*\*\*\*\*\*\*\*\*\*\*\*\*\*

**General Comment:** 1) *Introduction: The authors tried to emphasis the importance of lights based on the fact that power plants play significant role in air pollution. However, what the ratio of power electricity goes to lights? This information is not clear to me after reading the introduction. The same problem for the contents on Page 7, line 138-139.*

**Response**:

(1) To address the reviewer's comments, in the revised manuscript, we try to give more details regarding the ratio between power electricity and lighting electricity. In the present study, the potential emission reduction induce by the GLP was confined at the luminous efficacy, which is the core of the GLP. Between the base case (with the GLP) and sensitivity cases (without the GLP), the ratio of power electricity goes to lights is assumed same with the ratio of artificial lighting to the total electricity consumption, which is 10–14%. We added more explanation **in Line-273:** "... And the ratio of power electricity goes to lights is same with the ratio of artificial lighting to the total electricity consumption, which is 10–14% (Lv and Lv, 2012; Zheng et al., 2016)."

(2) The previous description is inaccurate in *line 138-139*, we revised the text **in Line-142:** "The rapid increase in the nighttime lights implicates that the lighting electricity were greatly increased."

**General Comment:** *2) Section 2.1. This section needs to be re-organized. The authors claim conclusions without showing data support. For instance, Page 7, line*

*154 and afterwards. "The above long-term variability of thermal power electricity and associated coal consumption for power generation was based on the situation that the GLP was conducted in China." The impact of GLP on power plants has not been quantified when stating so.*

**Response**:

(1) To address the comments of the reviewer, we clarify that the thermal power electricity and the total coal consumption of thermal power generation were based on the statistics of National Bureau of Statistics of China (NBS, 2000–2016). The air pollutant emission from thermal power generation respected to previous studies of *Liu et al., 2015 (Fig. 4)* and *Tong et al., 2018 (Fig. 2A)*.

(2) We change the sub-title of 2.1, "**2.1 The NTL and emissions from power sector**"

(3) By taking the comments of the reviewer, we have re-construct the ***Sect. 2.1***. The Section is re-organized except the start and end paragraphs. The rewritten text is highlighted in blue words in the revised manuscript.

**General Comment:** 3) *Page 12, line 264. "To estimate the emission reduction induced by the GLP, we assumed that the potential emission reduction was mainly due to the emissions from the thermal power plants." The uncertainty of this assumption is missing.*

**Response**: Thanks for the comments. We use some previous studies to estimate the uncertainties. For example, Thermal power generation is the primary electricity source in China, contributing to about 72–78% of the total electricity (NBS, 2000–2016), which indicates at least 6% uncertainty in the estimation. Lv and Lv (2012) and Zheng et al. (2016) estimate the ratio of artificial lighting to the total electricity consumption, and the ratio is 10–14%, which indicates about 4% uncertainty in the estimation. We revised the text **in Line-276:** "It is worth noting that, there were uncertainties in the present study. Thermal power generation is the primary electricity source in China, contributing to about 72–78% of the total electricity (NBS, 2000–2016), which indicates at least 6% uncertainty in the estimation. Lv and Lv (2012)

and Zheng et al. (2016) estimate the ratio of artificial lighting to the total electricity consumption, and the ratio is 10–14%, which indicates about 4% uncertainty in the estimation."

**General Comment:** 4) *Regional diversity of LED market share. The emission rates of power plants have large diversities over regions in China. If the LED market share has large variations over regions as well, the derived emission reductions will be significantly different from what was estimated in the current study.*

**Response**: We agreed the reviewer that the regional diversity of LED market share would significantly influence the derived emission reductions induced by the luminous efficacy improvement in the GLP. However, the lighting electricity is transported from the power plants. The spatial dynamics of derived emission reductions should be consistent with the distribution of power plants and the related coal consumption. The regional diversity of LED market share was finally included in the distribution of emissions from power sector, which respected to previous studies of MEIC in the present study (Zhang et al., 2009; Liu et al., 2015). We added more explanation **in Line-295:** "The regional diversity of LED market share would significantly influence the emission reductions derived by the luminous efficacy improvement induced by the GLP. However, the lighting electricity is transported from the power plants. The spatial dynamics of emission reductions induced by the GLP should be consistent with the distribution of power plants and the related coal consumption. The effects from regional diversity of LED market share was finally included in the distribution of emissions from power sector."

**General Comment:** 5) *Secondary particles in Table. The method of calculating secondary particles from power plants is not clear to me.*

**Response**:

(1) In Table 2, the X represents other air pollutant emissions (in addition to NOx, SO2, and PM$_{2.5}$) from power sector. Based on MEIC, the air pollutant emissions from

thermal power include many species, such as BC, OC, NOx, SO2, PM2.5, PMcoarse, VOCs, but not include secondary particles. We added more details **in Line-318:** "**Table 2** shows the emissions from power generation, including the NOx, $SO_2$, PM2.5 and other species (represented with X), such as the BC, PM coarse, VOC, and so on."

(2) The secondary particles from power plants is based on the WRF-CHEM model in ***Sect. 2.2* in Line-181:** "We also used a non-traditional secondary organic aerosol (SOA) model, including the volatility basis-set modeling approach and SOA contributions from glyoxal and methylglyoxal. Detailed information about the WRF-CHEM model can be found in previous studies (Li et al., 2010; Li et al., 2011a; Li et al., 2011b; Li et al., 2012)."

\*\*\*\*\*\*\*\*\*\*\*\*\*\*\*\*\*\*\*\*\*\*\*\*\*\*\*\*\*\*\*\*\*\*\*\*\*\*\*\*\*\*\*\*\*\*\*\*\*\*\*\*\*\*\*\*\*\*\*\*\*\*\*\*\*\*\*\*\*\*

**Specific Comment:** *1) abstract. "In the upper limit case of emission reduction. . ." and "in the lower limit case of emission reduction..." The English sounds not correct for me.*

**Response**: We revised the text **in Line 35-37 and Line 411-414**: "in the case of upper limit emission reduction..." and "in the case of lower limit emission reduction..."

**Specific Comment:** *2) line 77-78, please consider rephrasing "the famous nation-wide project of utilizing emission control facilities" . famous?*

**Response**: We deleted the "famous" **in Line-77**: "...by the nation-wide project of utilizing emission control facilities..."

**Specific Comment:** *3) line 88. Missing a space before 2005.*

**Response**: We revised the mistake **in Line-92**.

**Specific Comment:** *4) line 92-93. I don't quite get the meaning of the sentence. Please consider rephrasing it.*

**Response**: We revised the text to explain more clearly **in Line-95**: "Aside from emission reductions, the GLP is benefit to coal-saving from the thermal power generation, which inherently connected to air quality in China (Liu et al., 2015; Huang et al., 2016; Hu et al., 2016)."

**Specific Comment:** *5) line 105. The English is incorrect.*

**Response**: We revised the text to explain more clearly **in Line-109**: "There are several articles and books for summarizing the GLP from time to time by the Energy Research Institute under Chinas' National Development and Reform Commission, providing information for an assessment of the GLP."

**References**

Guo, F., and Pachauri, S.: China's Green Lights Program: A review and assessment, Energ. Policy, 110, 31-39, 2017.

Hu, J., Huang, L., Chen, M., He, G., and Zhang, H.: Impacts of power generation on air quality in China—Part II: Future scenarios, Resour. Conserv. Recy., 121, 115–127, 2016.

Huang, L., Hu, J., Chen, M., and Zhang, H.: Impacts of power generation on air quality in China—part I: An overview, Resour. Conserv. Recy., 2016.

Liu, F., Zhang, Q., Tong, D., Zheng, B., Li, M., Huo, H., and He, K. B.: High-resolution inventory of technologies, activities, and emissions of coal-fired power plants in China from 1990 to 2010, Atmos. Chem. Phys., 15, 18787-18837, 2015.

Li, G., Lei, W., Zavala, M., Volkamer, R., Dusanter, S., Stevens, P., and Molina, L.: Impacts of HONO sources on the photochemistry in Mexico City during the

MCMA-2006/MILAGO Campaign, Atmos. Chem. Phys., 10, 6551-6567, 2010.

Li, G., Bei, N., Tie, X., and Molina, L.: Aerosol effects on the photochemistry in Mexico City during MCMA-2006/MILAGRO campaign, Atmos. Chem. Phys., 11, 5169-5182, 2011a.

Li, G., Zavala, M., Lei, W., Tsimpidi, A., Karydis, V., Pandis, S. N., Canagaratna, M., and Molina, L.: Simulations of organic aerosol concentrations in Mexico City using the WRF-CHEM model during the MCMA-2006/MILAGRO campaign, Atmos. Chem. Phys., 11, 3789-3809, 2011b.

Li, G., Lei, W., Bei, N., and Molina, L.: Contribution of garbage burning to chloride and PM 2.5 in Mexico City, Atmos. Chem. Phys., 12, 8751-8761, 2012.

NBS, National Bureau of Statistics, China Statistical Yearbook 2000-2016, China Statistics Press, Beijing, available at: http://www.stats.gov.cn/tjsj/ndsj/

Tong, D., Zhang, Q., Liu, F., Geng, G., Zheng, Y., Xue, T., Hong, C., Wu, R., Qin, Y., Zhao, H., Yang, L., He, K., 2018. Current Emissions and Future Mitigation Pathways of Coal-Fired Power Plants in China from 2010 to 2030. Environmental Science & Technology 52, 12905-12914.

Zheng, B., Gao, F., and Guo, X.: Survey Analysis of Lighting Power Consumption in China, China Light & Lighting, 2016.